Largest global shark biomass found in the northern Galápagos Islands of Darwin and Wolf

Salinas de León Pelayo 1 pelayo.salinas@fcdarwin.org.ec
Acuña-Marrero David 1
Rastoin Etienne 1
Friedlander Alan M. 2 3
Donovan Mary K. 3
Sala Enric 2
1 Department of Marine Sciences, Charles Darwin Research Station , Puerto Ayora, Galapagos Islands , Ecuador
2 Pristine Seas, National Geographic Society , Washington, D.C. , USA
3 Fisheries Ecology Research Lab, University of Hawai’i at Manoa , Honolulu, HI , USA
Medina Mónica
Electronic publication date: 2016 May 10
Publication date: 2016
Volume: 4
Electronic Location ID: e1911
Received 2015 Oct 30; Accepted 2016 Mar 17
Copyright: ©2016 Salinas de León et al.
Copyright year: 2016
Copyright holder: Salinas de León et al.
License: This is an open access article distributed under the terms of the Creative Commons Attribution License, which permits unrestricted use, distribution, reproduction and adaptation in any medium and for any purpose provided that it is properly attributed. For attribution, the original author(s), title, publication source (PeerJ) and either DOI or URL of the article must be cited.
License URL: https://creativecommons.org/licenses/by/4.0/

Erratum in: Correction: Largest global shark biomass found in the northern Galápagos Islands of Darwin and Wolf 4 1 2 2018 e1911/correction-1 PeerJ PMC5794205 29404199
Keywords: Galapagos, Marine protected areas, Marine reserves, Sharks, Pelagics, Conservation, Biomass, Fish, Eastern Tropical Pacific, Darwin

Funding: Helmsley Charitable Trust Galapagos Conservation Trust Blancpain Davidoff Cool Water Funding provided by the Lindblad Expeditions-National Geographic Joint Fund for Conservation and Research, the Helmsley Charitable Trust, the Galapagos Conservation Trust, Blancpain, and Davidoff Cool Water. The funders had no role in study design, data collection and analysis, decision to publish, or preparation of the manuscript.

==============================
Overfishing has dramatically depleted sharks and other large predatory fishes worldwide except for a few remote and/or well-protected areas. The islands of Darwin and Wolf in the far north of the Galapagos Marine Reserve (GMR) are known for their large shark abundance, making them a global scuba diving and conservation hotspot. Here we report quantitative estimates of fish abundance at Darwin and Wolf over two consecutive years using stereo-video surveys, which reveal the largest reef fish biomass ever reported (17.5 t ha−1 on average), consisting largely of sharks. Despite this, the abundance of reef fishes around the GMR, such as groupers, has been severely reduced because of unsustainable fishing practices. Although Darwin and Wolf are within the GMR, they were not fully protected from fishing until March 2016. Given the ecological value and the economic importance of Darwin and Wolf for the dive tourism industry, the current protection should ensure the long-term conservation of this hotspot of unique global value.

Introduction

Overfishing has reduced biomass of most sharks and other large predatory fishes worldwide by over 90% (Baum et al., 2003; Myers & Worm, 2003; Ward-Paige et al., 2010), and even remote locations have been severely impacted (Dulvy et al., 2008; Sibaja-Cordero, 2008; Graham, Spalding & Sheppard, 2010; White et al., 2015). One in four species of cartilaginous fishes is now threatened with extinction due primarily to overexploitation and habitat loss (Dulvy et al., 2014). The systematic removal of sharks from marine ecosystems has negative effects that propagate throughout the entire food web (Bascompte, Melián & Sala, 2005; Myers et al., 2007; Heithaus, Wirsing & Dill, 2012).

Sharks and other top reef predators dominate healthy marine ecosystems, so that the traditional fish biomass pyramid is inverted in these increasingly rare areas (Friedlander & DeMartini, 2002; Sandin et al., 2008; Sandin & Zgliczynski, 2015). However, only a few localities worldwide still maintain large abundances of top predatory fishes due to either being remote and unfished, or having recovered after full protection from fishing (Sandin et al., 2008; Aburto-Oropeza et al., 2011; Graham & McClanahan, 2013; Friedlander et al., 2014a). The small number of scientific studies on relatively pristine ecosystems limits our ability to establish true baselines of sharks and other large predatory fish abundance and this restricts our capacity to determine realistic recovery targets for degraded ecosystems (McClenachan, Ferretti & Baum, 2012; Sala, 2015), thus perpetuating the shifting baseline syndrome (Pauly, 1995; Jackson, 2010). The establishment of marine protected areas (MPAs), especially no-take areas (NTA) where all forms of fishing are prohibited, have been shown to be one of the most successful management tools to confront global ecosystem degradation (Halpern & Warner, 2002; Lester et al., 2009; Edgar et al., 2014). A growing body of literature supports the positive effects of NTA, which includes substantial recoveries in fish abundance and biomass (Aburto-Oropeza et al., 2011; Eddy, Pande & Gardner, 2014); a greater biomass, abundance and size of top predators inside reserves than in nearby fished areas (see review by Lester et al., 2009); increase in abundance and biomass in nearby areas due to the spill-over of adults and/or larvae (Goñi et al., 2008; Halpern, Lester & Kellner, 2009; Christie et al., 2010), and shifts in species composition and trophic cascades that result in the restoration of entire ecosystems (Babcock et al., 1999; Babcock et al., 2010; Shears & Babcock, 2002; Shears & Babcock, 2003). Furthermore, a recent analysis across 87 sites globally revealed that conservation benefits of MPAs increase exponentially when reserves are no take, well enforced, old, large and isolated (Edgar et al., 2014).

The Galapagos Islands are known worldwide for its iconic terrestrial fauna and flora, due in large part to a young Charles Darwin who sailed to these islands in 1835 (Darwin, 1839). While Galapagos giant tortoises, Darwin’s finches, and mocking birds have received much of the attention since Darwin’s visit, the underwater Galapagos remains under-studied and largely unknown compared to terrestrial ecosystems. Galapagos is the only tropical archipelago in the world at the cross-roads of major current systems that bring both warm and cold waters. From the northeast, the Panama Current brings warm water; from the southeast the Peru current bring cold water; and from the west, the subsurface equatorial undercurrent (SEC) also bring cold water from the deep (Banks, 2002). The SEC collides with the Galapagos platform to the west of the Islands of Fernandina and Isabela, producing very productive upwelling systems that are the basis of a rich food web that supports cold water species in a tropical setting like the endemic Galapagos penguin (Spheniscus mendiculus) (Edgar et al., 2004). The oceanographic setting surrounding Galapagos results in a wide range of marine ecosystems and populations, that includes from tropical species like corals or reef sharks to temperate and sub-Antarctic species like the Galapagos fur seal (Arctocephalus galapagoensis) or the waived albatross (Phoebastria irrorata).

The far northern islands of Darwin and Wolf in the 138,000 km2 Galapagos Marine Reserve (GMR) represent a unique ‘hotspot’ for sharks and other pelagic species (Hearn et al., 2010; Hearn et al., 2014; Acuña-Marrero et al., 2014; Ketchum et al., 2014a). Most of the studies around this area have focused on the migration of scalloped hammerhead sharks (Sphyrna lewini) and other sharks species between Darwin and Wolf and other localities in the Eastern Tropical Pacific (Hearn et al., 2010; Bessudo et al., 2011; Ketchum et al., 2014a). An ecological monitoring program has visited the islands over the past 15 years with a strong sampling focus to survey reef fishes and invertebrate communities (Edgar et al., 2011). However, no study to date has examined extensively the abundance, size, and biomass of sharks and other large predatory fishes around Darwin and Wolf. We conducted two expeditions to Darwin and Wolf in November 2013 and August 2014 to establish comprehensive abundance estimates for shark and predatory fish assemblages at Darwin and Wolf. Our aim was to use this information to make recommendations for enhanced protection during the re-zoning process of the GMR started by the Galapagos National Park Directorate in 2015.

Materials and Methods

This research was approved by the Galapagos National Park Directorate (GNPD) as part of the 2013 and 2014 annual operational plan of the Charles Darwin Foundation.

Site description

Darwin and Wolf are the two northernmost islands in the Galapagos Archipelago, a group of 13 major islands and 100 islets and rocks located 1,000 km west of mainland Ecuador, in the ETP (Snell, Stone & Snell, 1996) (Fig. 1). The Galapagos Archipelago lies at the congruence of three major oceanic currents, which provides a highly dynamic and unique oceanographic settings (Palacios, 2004). Darwin and Wolf represent the far northern biogeographic region of the archipelago and are heavily influenced by the warm Panama current that comes from the Northeast, which supports sub-tropical marine communities to these islands (Edgar et al., 2004; Acuña-Marrero & Salinas-de-León, 2013). Darwin and Wolf are small (approximately 1 and 2 km2, respectively) and represent the tops of eroded, extinct submerged volcanoes, which rose from the surrounding seafloor >2,000 m below (McBirney & Williams, 1969; Peñaherrera-Palma, Harpp & Banks, 2013). Darwin and Wolf are exposed to a predominant north-western water flow that supports a unique pelagic assemblage on the south-eastern portions of these islands (Hearn et al., 2010). In contrast to much of the Galapagos, which is dominated by the cold equatorial counter-current, the waters of Darwin and Wolf range from 22.5 to 29 °C throughout the year, peaking during February–March (Banks, 2002). Two different seasons have been reported around Darwin and Wolf islands: a warm season from January to June, and a cool season from July to December, where mean sea surface temperature remains below 25 °C (Acuña-Marrero et al., 2014).

Figure 1 Location of Darwin and Wolf Islands within the Galapagos Marine Reserve, which encompasses the waters 40 nautical miles around the islands.

Black dots around Darwin (n = 3) and Wolf (n = 4) islands are survey sites. CDRS, Charles Darwin Research Station.

In this study, and for comparisons with other reefs worldwide, we treat Darwin and Wolf as a single ecological unit because of the following reasons. First, a number of published studies (Hearn et al., 2010; Ketchum et al., 2014a; Ketchum et al., 2014b) show that animals in Galápagos (especially hammerhead sharks) frequently move between islands during the cold season (July–December). Second, the two islands are a unique bioregion within the Galapagos Marine Reserve: the ‘far north’ (Edgar et al., 2004). These two islands are characterized by the influence of the tropical Panama current and support unique fish communities within the Galapagos Marine Reserve. Third, the islands are located less than 40 km apart, and isolated from the rest of the archipelago (Fig. 1).

Data collection

Underwater census using diver operated stereo-video

A diver operated stereo-video system (DOV) was used to sample fish assemblages around Darwin and Wolf over two consecutive years (2013, 2014) during the cold season that spans from July to December. DOVs use two Canon HFG-25 high-definition cameras mounted 0.7 m apart on a base bar inwardly converged at seven degrees and are operated by experienced divers using standard open-circuit SCUBA equipment. DOVs can overcome some of the biases associated with Underwater Visual Census (UVC) by eliminating the inter-observer effect and the over/underestimation of sampling area and fish lengths estimations (Harvey, Fletcher & Shortis, 2001; Harvey, Fletcher & Shortis, 2002; Harvey et al., 2003; Harvey et al., 2004; Goetze et al., 2015).

Fishes were surveyed at seven sites around Wolf (n = 4) and Darwin (n = 3) islands (Fig. 1) in November 2013 and August 2014. All sites were coastal rocky reefs and were selected based on their similar structure to be comparable. At each site, divers towed a surface buoy equipped with a GPS (Garmin GPSmap 78) to create a detailed track of the area surveyed, with GPS position and exact time recorded using a watch synchronized with the GPS at the beginning and end of each survey (Schories & Niedzwiedz, 2012). Divers followed the 20 m depth contour for a period of 25–30 min in order to complete a minimum of ten 50 m long and 5 m wide replicate transects at each site. Dive times were based on preliminary surveys that revealed that swimming at a constant speed, a 2-minute DOVS survey covered approximate 50 m. At some sites, strong currents resulted in longer distances covered by the survey team, resulting in a greater area surveyed. The diver towing the GPS also conducted standard UVCs to record sharks and large pelagics (50 × 5 × 5 m) in parallel to the stereo surveys, therefore the 2-minute surveys were also used to synchronize both sampling methodologies (Supplemental Information).

Calibration and video analysis

Stereo-video cameras were calibrated prior to field deployments using the program CAL (SeaGIS Pty Ltd; Harvey & Shortis, 1998). Following the dives, paired videos were viewed on a large monitor and analysed in the program Event Measure (SeaGIS Pty Ltd). Every fish observed was identified to species and measured to the nearest mm (Fork Length, FL). Lengths were converted to biomass (kg) using published length–weight relationships (Froese & Pauly, 2015). For individual fishes that were not measured (e.g., two individuals overlapping), we calculated biomass using an average total length for that species from the site where it occurred. Cryptic reef fishes (<8 cm) were excluded from our surveys due to the limited ability of the DOVs to detect these species and their lack of importance to the fisheries and overall biomass (Ackerman & Bellwood, 2000). Fishes were classified into four different trophic categories based on published information: apex predators, lower-level carnivores, planktivores and herbivores (Friedlander & DeMartini, 2002).

For largely abundant schooling fishes, primarily the abundant planktivorous species locally known as gringo (Paranthias colonus), which form dense schools that are difficult to quantify, we developed a specific methodology in the software Event Measure. For each of the study sites surveyed, we measured to the nearest mm a subsample of 100 individuals across all replicate transects and obtained a specific set of mean individual lengths. Then, transects were divided into blocks of identical length using the GPS tracks and every individual for each 10 × 5 × 5 m wide ‘cube’ was counted. The number of cubes varied according to transect lengths. Total biomass for these sites were obtained by multiplying the total numbers of individuals counted in each cube by the mean individual length for each species at that site.

We also conducted a comparative analysis between the traditional survey technique based on underwater visual censuses (UVC) and Diver Operated Video surveys (DOVs) using stereo-cameras to test for differences in estimates of shark diversity, abundance, and size (Text S1). A diver with >5 years experience in conducting visual surveys of sharks swam alongside a diver conducting video surveys. Both divers were synchronized to conduct the same transect in parallel. Synchronization was achieved by conducting 2-minute surveys. This time period was based on a previous archipelago-wide survey (n = 81 sites at 20 m) that showed that divers swimming at a constant speed covered an area of approximately 50 m during a 2-minute time period (P Salinas-de-León, 2014, unpublished data). A 15-second interval between transects was used to ensure independence between samples. The visual observer recorded individual shark species, size (FL), and sex of all sharks observed within a 5 m wide by 5 m high transect in front of the divers. Transect length was obtained by towing a GPS and dividing the GPS tracts into 2 min blocks, with a 15 s space between transects. We conducted a total of 69 transects across the seven study sites, covering a total area of 21,700 m2. Strong currents resulted in longer transects than previously estimated and mean transect length across study sites was 65.7 m (±2.2 SE). Transect length was not significantly different between sampling sites (ANOVA, p > 0.05).

Statistical tests

Patterns of total fish biomass and biomass without sharks between islands, wave exposures, and years were analyzed using generalized linear mixed models (Zuur, 2009) using the glmmADMB package (Skaug & Fournier, 2004) in the R statistical program version 3.0.2 (R Development Core Team, 2013). Due to the skewed nature of our biomass estimates, data were fit with a gamma error structure with an inverse link function that works well for continuous-positive data and has a flexible structure (Crawley, 2011). Islands, orientation, and year were all treated as fixed effects, while location (survey site) was used as a random effect in the model. Biomass by trophic group was assessed in a similar manner, except that data were fitted to negative binomial distributions due to the number of zero in these data. Unplanned post hoc multiple comparisons were tested using a Tukey’s honestly significant difference (HSD) test. Values in the results are means and one standard deviation of the mean unless otherwise stated. Comparisons between overall relative abundance and biomass recorded by UVC and DOVs were conducted using Wilcoxon rank sum tests.

Similarity of Percentages (SIMPER) in Primer 6.0 (Clarke & Gorley, 2006) was used to determine the fish species most responsible for the percentage dissimilarities between exposures using Bray-Curtis similarity analysis of hierarchical agglomerative group average clustering (Clarke, 1993). Differences in fish trophic biomass between islands, years, and wave exposures were tested using permutation-based multivariate analysis of variance (PERMANOVA, Primer v6.0, Clarke & Gorley, 2006). All factors and their interactions were treated as fixed effects. Trophic biomass data were 4th-root-transformed. Post hoc pair-wise tests were conducted between island, wave exposure, and year combinations. Interpretation of PERMANOVA results was aided using individual analysis of similarities (ANOSIM).

To describe the pattern of variation in fish trophic structure and their relationship to environmental factors we performed direct gradient analysis (redundancy analysis: RDA) using the ordination program CANOCO for Windows version 4.0 (TerBraak, 1994). Response data were compositional and had a gradient <3 SD units long, so linear methods were appropriate. The RDA introduces a series of explanatory (environmental) variables and resembles the model of multivariate multiple regression, allowing us to determine what linear combinations of these environmental variables determine the gradients. The environmental data matrix included island (Darwin, Wolf), wave exposure (NW, SE), and year (2013, 2014). To rank environmental variables in their importance for being associated with the structure of the assemblages, we used a forward selection where the statistical significance of each variable was judged by a Monte-Carlo permutation test (TerBraak & Verdonschot, 1995). Permutations tests were unrestricted with 499 permutations.

Results

Grand mean fish biomass between islands, years, orientation, and locations was 17.5 t ha−1 (±18.6 SE) and was 90% higher at Darwin (24.0 ± 20.8) compared with Wolf (12.6 ± 16.4), although this difference was not significant (Fig. 2 and Table 1). Biomass in the SE sections of both islands combined (26.9 ± 35.2) was more than 6 times higher than in the NW (4.4 ± 5.9). Biomass was similar between years (2013 = 19.3 ± 18.9; 2014 = 15.6 ± 19.5) but was significantly different due to the large year ×orientation owing to higher biomass in the NW in 2013 at both islands (Fig. 2 and Table 1).

Figure 2 Comparisons of total fish biomass by island, orientation, and year.

Box plots showing median (black line), mean (red line), upper and lower quartiles, and 5th and 95th percentiles.

Table 1 Comparisons of total fish biomass by island, orientation and year.

Results of generalized linear mixed models fit with a gamma error structure and an inverse link function. Unplanned post hoc multiple comparisons tested using a Tukey’s honestly significant difference (HSD) test. Only significant multiple comparisons are shown.

Factor	Estimate	Std. error	Z	P	Multiple comparisons	
Island	0.031	0.099	0.31	0.757		
Orientation	0.258	0.103	2.51	0.012*	SE > NW	
Year	0.435	0.117	3.72	<0.001***	2013 > 2014	
Orientation × year	0.449	0.116	3.88	<0.001***	SE13 = SE14 > NW13 > NW14	
Notes.

* p < 0.05.

*** p < 0.001.

Nearly 73% of the total biomass (12.4 ± 4.01 t ha−1) was accounted for by sharks, primarily hammerheads (Sphryna lewini—48.0%), Galapagos (Carcharhinus galapagensis— 19.4%), and blacktips (Carcharhinus limbatus—5.1%). Hammerheads occurred on 92% of transects at SE Darwin, 59% at SE Wolf, and 9% at both NW Darwin and Wolf. Gringos (Paranthias colonus) were the third most abundant species by weight, accounting for an additional 18.3% of the total biomass. They were 2.2 times more abundant by weight in 2013 (3.8 ± 4.1) compared with 2014 (1.7 ± 2.4). Gringos were 48% more abundant in the SE (3.5 ± 3.5) compared with the NW (2.4 ± 3.7) exposures. The average dissimilarity between orientations was 84%, with hammerhead sharks accounting for 41.6% of the dissimilarity, followed by gringos (24.2%), Galapagos sharks (12.8%), and blacktip sharks (3.4%) (Table 2).

Table 2 Fish species most responsible for the dissimilarity between northwest (NW) and southeast (SE) orientations based on Similarity of Percentages (SIMPER) analysis.

Species	SE	NW	Dissim.	% contrib.	Cumulative % contrib.	
Sphyrna lewini	15.06	0.7	35.0 (1.2)	41.6	41.6	
Paranthias colonus	3.55	2.4	20.3 (1.0)	24.2	65.8	
Carcharhinus galapagensis	4.66	0	10.8 (0.5)	12.8	78.6	
Carcharhinus limbatus	1.77	0	2.9 (0.2)	3.4	82.0	
Caranx melampygus	0.58	0.08	2.1 (0.3)	2.5	84.5	
Lutjanus argentiventris	0.31	0.07	1.3 (0.4)	1.5	86.0	
Lutjanus novemfasciatus	0.18	0.02	1.0 (0.3)	1.2	87.2	
Holacanthus passer	0.06	0.12	1.0 (0.3)	1.2	88.4	
Prionurus laticlavius	0.05	0.07	0.9 (0.4)	1.1	89.5	
Sufflamen verres	0.02	0.06	0.8 (0.3)	1.0	90.4	

Our comparison between underwater visual censuses (UVC) and diver operated video surveys (DOVS) showed that both methods recorded the same number of species (n = 4). Abundance of sharks recorded by DOVs was 1.18 ± 0.35 100 m−2 (mean ± SE), and 0.97 ± 0.29 ind. 100 m−2 by UVC, and they were not significantly different (W = 2,279; p-value = 0.619). Overall shark biomass recorded was not significantly different between methods (W = 2,341; p-value = 0.421), despite a 57% higher biomass recorded with DOVs (12.40 ± 4.01 t ha−1) compared to UVC (7.89 ± 2.05 t ha−1). DOVS yielded estimates of shark size significantly larger than visual surveys, which suggests that even experienced observers tend to underestimate shark lengths, particularly for the larger size classes (Fig. S1).

Fish biomass excluding sharks was 4.3 t ha−1 (±5.1), and was 68% higher at Darwin (5.8 ± 5.3) compared with Wolf (3.4 ± 4.8) but not significantly different between islands (Table 2). Exposure showed no significant difference in fish biomass without sharks, but was 58% higher at the SE (5.4 ± 5.3) compared to the NW (3.4 ± 4.7) exposures. Biomass without sharks was 67% higher in 2013 (5.2 ± 5.3) compared to 2014 (3.1 ± 4.5) but there was a significant interaction of year with wave exposure (Table 3).

Table 3 Comparisons of fish biomass without sharks by island, orientation and year.

Results of generalized linear mixed models fit with a gamma error structure and an inverse link function. Unplanned post hoc multiple comparisons tested using a Tukey’s honestly significant difference (HSD) test. Only significant multiple comparisons are shown. Exposure × year factors with the same letter are not significantly different (α = 0.05).

Factor	Estimate	Std. error	Z	P	Multiple comparisons	
Island	0.092	0.191	0.48	0.631		
Orientation	0.177	0.200	0.89	0.376		
Year	0.366	0.109	3.35	<0.001	13 > 14	
Orientation × year	0.281	0.109	2.58	0.009**	13SE 14NW 14SE 14NW A AB B C	

Apex predators (sharks, jacks, and groupers) accounted for 75% of the total biomass, followed by planktivores (primarily gringos) at 20%, lower level carnivores (4%), and herbivores (1%). Apex predator biomass was similar among years with a 27% difference (Table 4A). Darwin harbored apex predator biomass 2.7 times higher than Wolf, although these differences were not significant. Apex predator biomass was 24 times higher in the SE vs. NW orientations, and although results are suggestive, they were not significantly different owing to the high variance within orientations (NW COV = 271.6, SE COV = 155.5).

Table 4 Comparisons of biomass among trophic groups by island and orientation.

Results of generalized linear mixed models fit with negative binomial error structure. Unplanned post hoc multiple comparisons tested using a Tukey’s honestly significant difference (HSD) test. Only significant multiple comparisons are shown.

	Estimate	Std. error	Z	P	
A. Apex predators					
Island	0.673	0.692	0.97	0.33	
Orientation	3.071	1.731	1.77	0.07	
Year	0.001	0.650	0.01	0.99	
B. Planktivores					
Island	0.339	0.291	1.17	0.24	
Orientation	0.284	0.297	0.96	0.33	
Year	0.609	0.302	2.02	0.04*	
C. Carnivores					
Island	0.496	0.714	0.69	0.49	
Orientation	0.705	0.714	0.99	0.32	
Year	0.681	0.711	0.96	0.34	
D. Herbivores					
Island	0.435	1.022	0.43	0.67	
Orientation	0.662	0.992	0.67	0.50	
Year	0.427	0.941	0.45	0.65	
Notes.

* p < 0.05.

Biomass of planktivores was 2.5 times greater, and significantly so, in 2013 (4.5 ± 4.3) compared with 2014 (1.8 ± 2.4) (Table 4B). It was 79% higher at Darwin compared with Wolf and 29% higher in the SE vs. NW, although neither comparison was significant. Lower-level carnivores were 81% more abundant by weight in 2013 compared with 2014. Their biomass was 69% higher in the SE vs. NW, and 46% higher at Wolf compared with Darwin, although none of these comparisons were significant. Herbivore biomass was 97% higher in the NW vs. SE, 39% higher in 2014 vs. 2013, and 74% higher at Wolf compared with Darwin, but none of these factors was significant.

The structure of the fish assemblage at Darwin and Wolf based on the biomass of each trophic group was influenced by year, island, and orientation, as well as their interactions except for year × island (Table 5). Orientation (ANOSIM R = 0.39, p = 0.001), followed by year (R = 0.09, p = 0.001) appeared to have the strongest influence on trophic assemblage structure. Crossed ANOSIM between orientation and year yielded R = 0.41, p = 0.001 for orientation and R = 0.14, p = 0.001 for year. Crossed ANOSIM between orientation and island yielded R = 0.39, p = 0.001 for orientation and R = 0.04, p = 0.038 for island.

Table 5 PERMANOVA of drivers of the structure of fish assemblage at Darwin and Wolf based on the biomass of each trophic group (apex predators, planktivores, lower-level carnivores, and herbivores).

Only significant interactions are shown.

Source	df	SS	MS	Pseudo-F	P	
Year	1	17,232	17232.0	17.024	0.001	
Island	1	3,058	3058.4	3.022	0.027	
Orientation	1	54,298	54298.0	53.642	0.001	
Year × orientation	1	7,384	7383.9	7.295	0.001	
Island ×orientation	1	4,006	4005.5	3.957	0.010	
Year × island × orientation	1	4,345	4345.1	4.293	0.005	
Residual	129	130,580	1012.2			
Total	136	217,710				

Table 6 (A) Results of redundancy analysis (RDA) on square root transformed fish trophic biomass with environmental variables (e.g., island, orientation). (B) Conditional effects of Monte-Carlo permutation results on the redundancy analysis (RDA).

(A) Statistic	Axis 1	Axis 2	Axis 3	
Eigenvalues	0.34	0.04	0.01	
Pseudo-canonical correlation	0.67	0.46	0.20	
Explained variation (cumulative)	34.42	38.68	38.87	
Explained fitted variation (cumulative)	88.54	99.51	100.00	
(B) Variable	Pseudo-F	p	% explained	
Exposure	69.6	0.002	87.5	
Year	5.8	0.006	10.6	
Island	5.1	0.090	9.3	

The first two axes of the RDA bi-plot explained 39% of the functional group variance and 99% of the functional group-environment relationship (Table 6 and Fig. 3). Orientation explained 87.5% of the total variance, followed by year (10.6%), and island (9.3%). Orientations were well separated in ordination space with the SE orientation strongly correlated with apex predator biomass, while NW orientations were influenced by carnivore and herbivore biomass. Planktivore biomass was orthogonal to the other three trophic groups and drove the separation between years.

Figure 3 Bi-plot of results of redundancy analysis of fish trophic biomass with location, wave exposure, and year.

Blue circles represent the 2013 and 2014 data. Red triangles are orientations. Squares are centroids of Darwin and Wolf. Vectors are magnitude and directional effects of each trophic group on orientation of locations in ordination space.

Discussion

The first quantitative fish surveys using the stereo-video approach around Darwin and Wolf islands revealed the largest fish biomass reported to date on a reef worldwide (Fig. 4). This extraordinary biomass, which consists mostly of sharks, is considerably larger than that reported at Cocos Island National Park (Friedlander et al., 2012) and the Chagos Marine Reserve (Graham et al., 2013), the next largest fish biomasses globally.

Figure 4 Biomass at Darwin and Wolf compared to other remote locations and no-take marine reserves around the world (mean ± SE).

Data from DeMartini et al., (2008); Sandin et al., (2008); Aburto-Oropeza et al., (2011); Friedlander et al., (2012); Friedlander et al., (2013); Friedlander et al., (2014a); Friedlander et al., (2014b); Graham et al., (2013).

One potential issue is the comparison of shark biomass results from different methods. We obtained biomass estimates using two methods: underwater visual censuses and video censuses, and found no significant differences between the two methods, despite a 57% difference in mean biomass between them. This difference is probably due to the large variance in shark biomass values, the small sample size, and the underestimation of shark sizes by visual censuses (Fig. S1). There is no practical way to deal with the issue of small sample size since Darwin and Wolf are small islands, and the availability of comparable sampling habitat is limited. The issue of large variance is also endemic to areas that show large concentrations of top predators in specific spots, typically exposed and with strong currents (Friedlander et al., 2012; García-Charton et al., 2004; Hearn et al., 2010), making the distribution of top predators highly heterogeneous. In any case, even if the 57% difference between methods were significant, the shark biomass at Darwin and Wolf (12.4 t ha−1) would still be greater than any other location globally (Cocos Island would be the closest with a re-estimated upper value 7.1 t ha−1 for apex predators).

Our results contribute to the growing body of literature that demonstrates that the least impacted areas are dominated by top predatory fishes, mainly sharks (Friedlander & DeMartini, 2002; Sandin et al., 2008; Graham et al., 2013; Friedlander et al., 2013; Friedlander et al., 2014a). At Darwin and Wolf, top predators account for an astonishing 75% of the fish biomass. Inverted fish biomass pyramids had been unreported until recent surveys of unfished coral reefs (Sala, 2015). Such inverted biomass pyramids of subsets of ecological communities can be maintained when the top levels of the food web have a much lower turnover rate (slower growth rate per biomass unit) than their prey (Sandin & Zgliczynski, 2015), and/or when they are subsidized by external energy inputs (Trebilco et al., 2013). In the case of Darwin and Wolf, the large predatory fish biomass might be supported not only by the large abundance of lower trophic levels fish on the reefs but also by the very productive surrounding pelagic waters, where hammerhead and other sharks take daily foraging excursions (Ketchum et al., 2014a; Ketchum et al., 2014b).

Figure 5 Common encounters around Darwin and Wolf Islands.

(A) A large school of hammerhead sharks (Sphyrna lewini); (B) A group of Galapagos sharks (Carcharhinus galapagensis), including a couple of pregnant females; (C) A large female whale shark (Rhincodon typus) swims among a school of hammerhead sharks. All photos by Pelayo Salinas-de-León.

Sharks, mainly hammerhead and Galapagos sharks, dominated the fish assemblage, but other predators like the bluefin trevally, black jack (Caranx lugubris) and bigeye jack (C. sexfasciatus) were also common at several of the sites surveyed (Figs. 5A–5C). Our results revealed a marked concentration of sharks and planktivorous fish biomass at the southeast corners of Darwin and Wolf, something previously documented by acoustic telemetry studies (Hearn et al., 2010; Ketchum et al., 2014b). The higher fish abundance at these SE locations may be related to local oceanographic features, dominated by a unidirectional current from the southeast to the northwest that collides with the southeast corner of both islands (Hearn et al., 2010). This current may enhance productivity that supports rich benthic communities and large numbers of planktivorous fishes, mainly gringos, which may serve as a food source to carnivorous fishes and sharks (Hamner et al., 1988; Hearn et al., 2010). Other proposed hypothesis, include that this area constitutes a vantage location for nightly foraging excursion to adjacent pelagic areas; and/or this area is an important cleaning station (Hearn et al., 2010; Acuña-Marrero et al., 2014; Ketchum et al., 2014b). It is important to consider that these results likely represent maximum annual shark biomass because the surveys were carried out during the cold season (July–December), when hammerhead and other sharks are most abundant (Palacios, 2004; Hearn et al., 2014; Acuña-Marrero et al., 2014; Ketchum et al., 2014b). Seasonal changes in fish assemblages and biomass are likely since hammerheads are known to migrate from these islands between February and June (Ketchum et al., 2014b). Future studies should focus on seasonal trends and depth gradients (Lindfield, McIlwain & Harvey, 2014) in shark abundance and distribution.

A total ban on the capture, transport, and trade of sharks within the GMR was established in 2000 (AIM, 2000). However, illegal fishing within GMR boundaries (Jacquet et al., 2008; Carr et al., 2013) and recent efforts by local artisanal fishermen to expand longline fishing, a practice banned since 2005 due to large by-catch (Murillo et al., 2004), threaten shark populations. While veteran divers report larger abundance of sharks at Darwin and Wolf 30 years ago (Peñaherrera-Palma et al., 2015), the absence of long-term quantitative studies to monitor shark and large pelagic fish on a systematic basis and with enough replication does not allow an accurate assessment of the magnitude of decline of shark populations at Darwin and Wolf. By comparison, the recent analysis of a 21-year monitoring program for sharks and large pelagic fishes at Cocos Island National Park in Costa Rica revealed a sharp decline in 8 of the 12 elasmobranch species monitored, including the endangered hammerhead shark and the giant manta ray (Manta birostris) (White et al., 2015).

Despite the large shark biomass at Darwin and Wolf, our surveys also revealed a low overall biomass of predatory reef fishes such as the leatherbass (Dermatolepis dermatolepis) and the sailfin grouper (Mycteroperca olfax), both endemic to the Eastern Tropical Pacific (ETP) (Grove & Lavenberg, 1997). These species are highly prized by Galapagos artisanal fishermen, but their life histories (e.g., long lives, slow growing) make them extremely vulnerable to overfishing (Aburto-Oropeza & Hull, 2008; Usseglio et al., 2015). Leatherbass biomass reported for Darwin and Wolf (0.008 t ha−1 ± 0.05 SD) is 14 times lower than at the unfished Cocos Island (0.1 t ha−1) (Friedlander et al., 2012). Artisanal fishermen are known to directly target the only reported spawning aggregation for M. olfax in the GMR (Salinas-de-León, Rastoin & Acuña-Marrero, 2015), an unsustainable fishing practice known to deplete reefs fish stocks at an alarming rate (Sala, Ballesteros & Starr, 2001; Sadovy & Domeier, 2005; Erisman et al., 2011; Hamilton et al., 2012). The low biomass estimates for groupers reported here are likely caused by the unregulated artisanal fishery for demersal fishes in the GMR that directly targets over 50 coastal fish species and has been shown to have a negative impact on coastal resources of the GMR (Ruttenberg, 2001; Molina et al., 2004; Burbano et al., 2014; Schiller et al., 2014).

Our results also add to the growing body of literature that supports the use of the stereo video methodology as a complement to traditional visual census, as this technique improves the accuracy and precision of fish length estimates (Harvey, Fletcher & Shortis, 2001; Harvey, Fletcher & Shortis, 2002), produces more accurate estimates of area surveyed (Harvey et al., 2004), and eliminates the inter-observer bias associated with species identification (Mallet & Pelletier, 2014). Although both stereo-DOVS and UVC recorded a similar number of shark species and overall relative abundance, in our study (one of the few to evaluate the use of DOVs with large and highly mobile species such as sharks), confirmed that even experienced divers tend to underestimate the individual length of large fishes.

Figure 6 High-resolution bathymetry around Darwin and Wolf Islands.

Recent multi-beam echo sounder surveys around D&W have revealed the presence of a number of seamounts (white triangles) and active hydrothermal vents and black smokers (white stars) that support unique biological communities. Additional inferred seamounts (grey triangles) are likely to be discovered to the West of the Islands. Source: Dennis et al., (2012), Ocean Exploration Trust NA-064 2015. Map Data: Google Earth, Data SIO, NOAA, US Navy, NGA, GEBCO, Data LDEO-Columbia, NSF, NOAA.

Conservation implications

This study adds to the growing body of literature that highlights the ecological uniqueness and the global irreplaceable value of Darwin and Wolf (Salinas-De-León et al., 2015). These islands not only harbour the largest shark biomass reported to date, but also represent a unique tropical bioregion within the GMR (Edgar et al., 2004). In addition, they are home to the last true coral reefs in the GMR (Banks, Vera & Chiriboga, 2009; Glynn et al., 2009). These islands also represent essential stepping stones for endangered and highly migratory species, such as hammerhead sharks (Hearn et al., 2010; Bessudo et al., 2011; Ketchum et al., 2014a). They are key waypoints for a recently documented migration probably related to reproductive purposes for the largest fish species on the planet, the whale shark Rhincodon typus (Acuña-Marrero et al., 2014) (Fig. 5D), and are home to the only known reproductive aggregation for the regionally endemic sailfin grouper (Salinas-de-León, Rastoin & Acuña-Marrero, 2015). These islands are visited by deep-water species such as the smalltooth sandtiger shark Odontaspis ferox (Acuña-Marrero et al., 2013), and are surrounded by numerous seamounts and active hydrothermal vents that harbour unique biological communities (P Salinas-de-León, 2014, unpublished data) (Fig. 6).

The economic benefits of ecotourism from sharks are far greater than shark fishing (Clua et al., 2011; Gallagher & Hammerschlag, 2011; Cisneros-Montemayor et al., 2013). For instance, the net present value of the average hammerhead shark at Cocos Island National Park was estimated at $1.6 million, compared to the ∼$200 that a fisherman obtains by selling a dead shark (Friedlander et al., 2012). In Galapagos, the net present value of a shark to the tourism industry is an astonishing $5.4 million (Lynham et al., 2015). The value of an individual shark to the tourism industry is ∼$360,000 per year, compared to $158 obtained from a dead shark (Lynham et al., 2015). That makes sharks alive in Galapagos the most valuable on Earth. Despite their high economic value and iconic importance, only about 50 km2 of the waters around Darwin and Wolf (representing an insignificant 0.04% of the total GMR area) were fully protected from fishing after the creation of the GMR in 1998.

Given the large-scale migrations reported for several shark species around Wolf and Darwin (Bessudo et al., 2011; Ketchum et al., 2014a), and the night foraging excursions by scalloped hammerhead sharks of up to ∼30 km from shore (Ketchum et al., 2014a), the levels of protection after the creation of the Galapagos Marine Reserve in 1998 were clearly insufficient. The government of Ecuador created a 40,000 km2 no-take reserve (the ‘Darwin and Wolf Marine Sanctuary’) in March 2016, expanding levels of protection around Darwin and Wolf, including some of the numerous seamounts located around these islands (Fig. 6). This conservation move is critical to ensure the recovery and long-term preservation of one of the most extraordinary marine ecosystems on the planet—and an economic engine for Ecuador.

Supplemental Information

Supplemental Information 1 Fish biomass 2014, 2015

Click here for additional data file.

Supplemental Information 2 Fish biomass 2014, 2015 without sharks

Click here for additional data file.

Supplemental Information 3 Supplementary Information file

Click here for additional data file.

We would like to thank the Charles Darwin Foundation and the Galapagos National Park Directorate for their institutional support to this study. We are grateful to all the staff of the Charles Darwin Research Station that made this study possible. Special thanks to Jennifer Suarez and Jose Feijoo for their field assistance. We are also very grateful to the crew of the MV Queen Mabel expedition yacht for their logistical support. This publication is contribution number 2135 of the Charles Darwin Foundation for the Galapagos Islands.

Additional Information and Declarations

Competing Interests

Author Contributions

Field Study Permissions

Data Availability

The authors declare there are no competing interests.

Pelayo Salinas de León, David Acuña-Marrero and Etienne Rastoin conceived and designed the experiments, performed the experiments, analyzed the data, contributed reagents/materials/analysis tools, wrote the paper, prepared figures and/or tables, reviewed drafts of the paper.

Alan M. Friedlander and Enric Sala conceived and designed the experiments, analyzed the data, contributed reagents/materials/analysis tools, wrote the paper, prepared figures and/or tables, reviewed drafts of the paper.

Mary K. Donovan analyzed the data, wrote the paper, prepared figures and/or tables, reviewed drafts of the paper.

The following information was supplied relating to field study approvals (i.e., approving body and any reference numbers):

This project was conducted under research permits granted to the Charles Darwin Foundation from the Galapagos National Park Directorate.

The following information was supplied regarding data availability:

Data can be found in the Supplemental Information.

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
