# Peer review of "Largest global shark biomass found in the northern Galápagos Islands of Darwin and Wolf"

_PeerJ, doi:10.7717/peerj.1911_

## Round 0.1 · original submission · Major Revisions

Three reviewers agree that this manuscript has potential for being a valuable contribution but they point out major issues primarily related to the statistical analysis and the interpretation of your results. The three reviews are thoughtful and thorough. Please address their comments carefully for resubmission.

·

Basic reporting

OVERVIEW

In this manuscript the authors provide estimates of the reef fish biomass at two remote Islands part of the Galápagos Archipelago. The authors also provide comparisons of the trophic structure between islands and “exposure”. Data was mostly obtained through stereo-video surveys but also include a comparison with visual census conducted by the diver. Overall the authors present valuable data from understudied habitats with unique conditions that support impressive fish biomass, specially sharks. This could be an interesting contrast against other areas under more intense human impact. Even though I am supportive of the publication of this data and believe that PEERJ is the most appropriate Journal based on its contents and broad readership, the current version of the manuscript lacks polish and I believe it needs major revision before it is considered for publication. Please see specific comments in the annotated PDF that I hope will be useful to improve this manuscript. I would like to see the revised version of this manuscript.

MAJOR POINTS:

- I believe the introduction is too simple and fail to frame the study into a broader scale. This can be easily fixed by adding more paragraphs that develop more the idea of pristine environments as well as the productivity in the Galapagos region.

- The manuscript would greatly benefit from this literature review, followed by some adjustments in the discussion to make it more appealing to a broader readership.

- Data analysis is not sufficiently or clearly explained in the Methods which jeopardize the interpretation of the results presented by the authors. Particularly, I suggest the authors to clarify the PERMANOVA design and results.

MINOR POINTS:

- By the end of the cover letter, authors state: “We believe this manuscript will be of interest to the Plos One readership”; and on line 250 the references are cited according to the PLoS One format. Although these are simple mistakes, they demonstrate that the article still needs polishing before it can be considered for publication.

(Please see more detailed comments in the annotated PDF file)

Experimental design

- Field work and data collection were appropriate to a descriptive study, particularly considering that it was performed in such isolated habitats. However, data analysis is not clearly explained and may be misleading as it jeopardizes the interpretation of the results. I am confident this can be addressed in a new version of the manuscript, but in my opinion it is not satisfactory in this current version.

Validity of the findings

- The authors have a good data set but the lack of clarity of statistical analysis which puts come of the results in question. Once this is addressed, I believe the results could be of interest to a wide readership.
- The authors could also state more clearly what was the major outcome of this paper. (please see more comments in the annotated PDF).

Reviewer 2 ·

Basic reporting

see general comments

Experimental design

see general comments

Validity of the findings

see general comments

Additional comments

The manuscript entitled "Largest global shark biomass found in the northern Galápagos Islands of Darwin and Wolf" by Leon et al. suffers from an apparent spread disease in the conservation literature: starting from a good dataset the authors too quickly seem to forget the science side of things and move on to the political side. The penalty is clear.

In first place I want to make it clear that I do believe the authors have a very interesting dataset, despite apparent limitations on spatial coverage. There are, however, some key issues that preclude this paper from being published as it is. The main "finding" of the present manuscript, and the one that grounds it is that shark biomass on the islands of Darwin and Wolf in the Galapagos Archipelago is the largest ever reported. This statement is based on a multi-study, multi-method comparison of regionally estimated "magical" biomass values. Strong inconsistencies and inadequacies are discussed below:

1) Political reasoning trampling over science: according to the authors, biomass estimate in Darwin and Wolf Islands is 15.9 t/ ha, a really high value. But when they spatially decompose values from the two islands we see a huge heterogeneity: Darwin "with" 28.3 and Wolf "with" 6.3 t/ha! Further on in their regional graph comparing measured biomass in the present study with estimates from other places around the world the authors insert the mean value of the two islands. What does this mean means? If we are to put the two islands separately, Wolf would be somewhere near Kingman and not "that" special anymore. So what do the authors do? Nothing. They simply ignore this fact and go on considering both islands as an entity because they already know what they pledge: to fully protect these islands from fishing. No matter what they have really measured. This is a valid pledge, with which I completely agree. It should however be done without ignoring their own data, but rather by discussing the reasons behind it.

2) Different sampling methods among studies. The present study estimated biomass through diver operated stereo video-system. All other studies´ with which they compare are derived from Underwater Visual Census. The authors simply assume both methods are comparable and go on as if there was nothing to worry about. There is a brief mention at the methods that the authors made also UVCs paired with DOVs. This is not mentioned anywhere else in the text so that we wonder whether they really did both methods and if they bothered comparing between them. Then I looked at the Sup. Mat. and discovered the comparison was there! Why is it not at the main text? Did the authors think of it as unnecessary? I think this is an essential part since all of their results sit on the assumption that the methods are comparable, at least for estimating biomass of large-sized or large-schooling fishes.

The next question is: are they really comparable? At the Supl. Material the authors only make limited comparisons on absolute estimates of richness, abundance and biomass of each method (and a comparison on estimates of shark size classes). The comparison on biomass reads:
"Similarily, overall relative biomass recorded was not significantly different between methods (W=2341; p-value= 0.421), despite a 57% higher biomass recorded with DOVs (12.40 ± 4.01 t ha-1) compared to UVC (7.89 ± 2.05 t ha-1)."

So the authors accept a 57% of difference between methods as comparable just because of statistical "insignificance"? This is just a result of high variability of the data and small sampling size, not of absence of effect! I am pretty sure if there were more samples, very high differences would be found. Possibly if comparisons were made by site, variability reduction would also allow for differences to be detected. If not, randomization methods can certainly work this out.

And then, once the authors show, by their data, that biomass estimates are not directly comparable they will pretty much invalidate their own approach. Probably the only way out is to derive a correction index for their estimates. One thing to note is that if the 57% overestimation stands, "biomass" at Darwin and Wolf would be down to ~6.8 t/ha both, with Darwin going to ~12.2 t/ha and Wolf to ~2.7 t/ha. This is a picture VERY different from the one the authors tried to portrait.

3) As it is becoming classic in large-scale comparisons of biomass (many of them led by some of the authors here), values for regional comparison are reported without any measure of data dispersion. This is the most basic statistic principle and it is not worth going over it. I known that it is not the authors´ intent to statistically test differences. But we known that biomass data is largely overdispersed (the authors recognize this when they try a glmm with gamma distribution) with a lot of extreme values that heavily influence the average. This is due to extreme spatial, temporal and habitat heterogeneity. The point here is: what is the meaning of comparing average biomass from different regional studies in different spatial scales (number of sites sampled, number of stations within a site) and distinct habitats? What is the philosophical point in stating "biomass in place A is X ton/ha"? It is much more informative to say that "biomass in place A varied from X to Y, with concentrated values close to X but some biologically meaningful values closer to Y". Representing standard error of the mean in a bar chart is a consequence of these inconsistencies and a really effective way of hiding the intrinsic data variability.

4) Estimating shark biomass from non-instantaneous counts. The authors ignore the problems with estimating the biomass of highly mobile fishes (such as sharks) from non-instantaneous counts (Ward-Paige et al. 2010. Overestimating fish counts by non-instantaneous visual censuses. PLoS ONE e11722).

In addition to these critical issues, there are some smaller points which also undermine the quality of the manuscript:

Use of inadequate statistical analysis: the authors employ a Redundancy Analysis with two categorical variables. Albeit it seems there is no mathematical drawback from this, I cannot see any reason for such an approach since a simpler MDS or PCoA plot with colored groups (the same categories the authors employ as "environmental variables") can answer the same questions. To worsen things, one variable ("exposure", in fact the direction where surge predominantly comes) is nested within the other. Following the wave of studies employing mixed-modeling, the authors go on with this approach also to test differences between sites. They make a good choice for the generalized model part employing a gamma distribution (adequate for biomass). But then they use a random factor with 2 levels? Please see Brian McGill´s important post with comments on this deadly sin (https://dynamicecology.wordpress.com/2015/11/04/is-it-a-fixed-or-random-effect/).

Figures: data presentation is based on bar charts, an effective way of hiding your data. But ecologists are starting to move on to a new data presentation paradigm. See "Kick the bar chart habit" Nature Editorial Vol 11, N° 2, Feb-2014 and Weissgerber et al. (2015) Beyond Bar and Line Graphs: Time for a New Data Presentation Paradigm. PLoS Biology. Moreover Figure 1 is a "frankenstein" figure, very effective in making the reader confused. There is no ordination between panels. There are no geographical indications, nor even directions. If I did not known where Galapagos and, particularly Darwin and Wolf are, I could NEVER realize based on this figure. It is also completely impossible to assert which island is Darwin and which is Wolf based solely on it. The caption is also useless.

Together all this problems result in a paper impossible to be accepted in its current form. I recognize, however, that there is potential on the data if the authors are willing to rework the manuscript and resubmit it.

·

Basic reporting

Lines 1, 12, 307and 310: "Galápagos" with no accent mark in English: "Galapagos".
Line 23: change authors' order (alphabetically).
Line 42: check for the missing word: "years".
Line 54: check for unnecessary initials in the reference: "Snell, Stone & Snell, H. L., 1996".
Line 55: consider "unique oceanographic settings" instead of "setting", since they are changing conditions (several of them).
Lines 2, 59 and 246: if both are the same author, unify the surname: "Salinas-De-León" vs. "Salinas-de-León".
Line 65: grad symbol is missing: "29 °C".
Line 76: a period is missing at the end of the sentence.
Line 90: the map in figure 1 lacks labels. Which one is Darwin and which one is Wolf? From the number of surveys conducted that information might be inferred but labels should be inside the figure. Also for the ETP and at least continental Ecuador.
Line 93: a coma is missing between the authors and the year of their publication.
Line 101: Authors mean "trophic" instead of "tropic"?
Line 102: Line 76: a period is missing at the end of the sentence.
Line 119: the word "that" is missing: "except data" vs. "except that data".
Lines 124 and 131: unify the surname "Ter Braak". Also in the reference section (line 366) place "Ter Braak" in the "T-section" instead of the "B-section". The surname first letter is "T" not "B".
Line 145: the word "diff" looks incomplete. Should it be "differ" or "differed"?
Line 222: a period is missing at the end of the sentence.
Line 242: Letters A, B and C are not in the corresponding pictures of the figure.
Lines 226-228: this sentence belongs to Results.
Line 250: check for extra spaces between "due to extremely".

Experimental design

Line 147: Authors should address some general details about the sampling sites in the site description section. Locations seem to differ greatly, thus differences in biomasses might be related to those differences between sites.

Validity of the findings

Line 184: Avoid the use of such adjectives to refer to a difference within the same order of magnitude; also, consider that the SD are greater than the average biomass value (thus, differences should be treated with care).

Results:
1-How can the Authors be sure that the individuals they saw in the different locations are not the same? In other words, how can the Authors be sure that they are not overestimating numbers (and thus biomasses) of highly mobile species as sharks are? In the Excel file there are no dates for the records, but it looks like the data were not taken simultaneously (but at different days).
2-Considering that all the data presented correspond to a single month, all comparisons between sites, exposures, oceanographic conditions are highly speculative. Too much attention is placed to those comparisons, which should be performed with at least a series of data from a whole year (including all oceanographic conditions, which fluctuate). Giving so much attention to those circumstantial comparisons deviates the attention of the reader from the main result of the study: the high biomass recorded and the need for conservation measurements.

Part of the discussion and the main conclusion of the study (and the most relevant result) must be presented in an improved manner:

1-"the largest fish biomass reported to date worldwide" must be presented in relation to shore or island systems, because there are well documented fish biomasses greater than the one found in Darwin and Wolf, thus, it is not the largest worldwide: it might be the largest documented on shores or island systems (coral and rocky reef habitats). For example, schools of some tuna species exceed the biomass found by the authors in Darwin and Wolf. They are also apex predators, commonly traveling in open ocean waters, along with other apex predators associated to those tuna schools (i.e., sharks and dolphins).

2-in this first part of the discussion the theory of the "Inverted biomass pyramid" is presented as the possible explanation of the great biomass supported by the shallow (and adjacent) systems of Darwin and Wolf, suggesting that this higher-than-usual biomass "can be maintained when the top levels of the food web have a much lower turnover rate (slower growth rate per biomass unit) than their prey" (quoting Sandin & Zgliczynski, 2015). This explanation seems unlikely. Such "inverted pyramids" are usually observed when the system is subsidized by other (sometimes adjacent) system(s). In the case of Darwin and Wolf, the ecology of the most abundant species gives clues on a more plausible explanation. Sphyrna lewini is a very mobile shark, which seems to spend the day close to islands (shallow systems) while moving to deeper waters in search for food during the night. There have been records of individuals from this species moving from one island (i.e., Malpelo) to another (i.e., Galapagos), sometimes covering distances of up to 1000 km in a few days (one paper with such data quoted by the Authors). In the absence of philopatric data, and with just one field trip of data, Authors should be careful when trying to explain the great biomass found. Replicates during at least a whole year should reveal how "resident" is the hammerhead sharks' population and, thus, how dependent on the local preys the hammerhead sharks are indeed.

Lines 233-235: this sentence is highly speculative, considering that the Authors also say that hammerhead sharks (most abundant species = highest biomass) is not a "resident" species. The sentence is implying that those sharks feed on gringos and other local "resident" species, but this is not supported by the data presented (also see the comment on the trophic ecology of this shark above).

Lines 309-310: Authors should consider rephrasing the sentence "That makes sharks alive in Galápagos the most valuable on Earth.", because this conclusion arises from data that were not intended to estimate an economic value. Moreover, as hammerhead sharks migrate and leave Darwin and Wolf (actually they might leave the entire GMR for several months) they should be consider as a "moving treasure" in terms of their economic value. If they, for example, migrate to Cocos or to Malpelo, they should be seen as a shared good (as all migratory species are; they don´t belong to a particular country).

Additional comments

See comments on the attached file.

---

## Round 0.2 · Minor Revisions

Thank you for your careful revision. This is a much improved version of the manuscript. The reviewers have provided positive feedback but also point out some biases in your discussion. Please make sure to address the comments regarding biomass underestimation.

Reviewer 2 ·

Basic reporting

Reviewing the manuscript entitled "Largest global shark biomass found in the northern Galápagos Islands of Darwin and Wolf" I have found that overall it has considerably improved since the previous version. The authors might have been somewhat upset with my comments and might disagree with some of the considerations, but the outcome of the reviewing process is undeniably positive. The idea has always been to draw the attention to controversial, almost "taboo" topics in the reef fish community literature. Some drawbacks in their work, however, are still present.

1) The authors insist on disregarding the considerable, albeit not "significant" differences on biomass estimates between UVC and DOV. The amount of underestimation of UVC has to be considered, both in the text and in Figure 4. Using their own calculated "correction factor" would be a practical way of addressing this. I agree with them that now, in face of more data, it will not change overall conclusions (i.e. Darwin and Wolf have the largest shark biomass reported to date on any given tropical reef). However it is still needed.
2) The authors insist on disregarding the fierce critics to estimates of shark abundance based on UVC (Ward-Paige et al. 2010 PLoS One). I do not pledged the abandonment of the method. But it is crucial for the author to explicitly address this issue on the text. It would be an elegant way of dealing with this undeniable problem.
3) In the Introduction (lines 41-43), authors should mention the theory behind Inverted Biomass Pyramids (see Trebilco et al. 2013 Ecosystem ecology: size-based constraints on the pyramids of life. TREE). Also, since no study has measured biomass of all living compartments in these pristine areas (only of reef fishes), authors should be more careful when asserting that these areas' biomass pyramids are inverted. This is their hypothesis, and only limited amount of data (fish surveys) exist to support it.
4) Again on this issue, in the Discussion (lines 303-306), as discussed in Trebilco et al. 2013 TREE (cited above): higher turnover is not required to explain an inverted biomass pyramid. Instead, large differences in the mean size of predators and prey and energetic subsidies from adjacent productive habitats should account for it. It is important for the authors to discuss these ideas at this point
5) The authors should include in the methods section the biogeographical and oceanographic reasons for grouping Darwin and Wolf at the global comparisons. This was cited in the answer to reviewer's comments.

Please see more comments and suggestions on the submission pdf attached.

Experimental design

"No Comments"

Validity of the findings

"No Comments"

Additional comments

"No Comments"

Annotated reviews are not available for download in order to protect the identity of reviewers who chose to remain anonymous.

·

Basic reporting

See all my final comments in the attached PDF-file (the 1st 2 pages contain my review, the remaining pages are the 'Reviewing PDF' where I have highlighted, in yellow, the text which should be changed.

Experimental design

See all my final comments in the attached PDF-file (the 1st 2 pages contain my review, the remaining pages are the 'Reviewing PDF' where I have highlighted, in yellow, the text which should be changed.

Validity of the findings

See all my final comments in the attached PDF-file (the 1st 2 pages contain my review, the remaining pages are the 'Reviewing PDF' where I have highlighted, in yellow, the text which should be changed.

Additional comments

See all my final comments in the attached PDF-file (the 1st 2 pages contain my review, the remaining pages are the 'Reviewing PDF' where I have highlighted, in yellow, the text which should be changed.

---

## Round 0.3 · accepted · Accept

Congratulations on a valuable contribution.